# Deep Reinforcement Learning for Organ Localization in CT

**Fernando Navarro**[1]                                                                 FERNANDO.NAVARRO@TUM.DE

**Anjany Sekuboyina**[1]                                                             ANJANY.SEKUBOYINA@TUM.DE

**Diana Waldmannstetter**[1]                                             DIANA.WALDMANNSTETTER@TUM.DE

**Jan C. Peeken**[2]                                                                              JAN.PEEKEN@TUM.DE

**Stephanie E. Combs**[2]                                                          STEPHANIE.COMBS@TUM.DE

**Bjoern H. Menze**[1]                                                                        BJOERN.MENZE@TUM.DE

[1] *Department of Informatics And Mathematics, Technical University of Munich, Germany*

[2] *Department of Radio Oncology and Radiation Therapy, Klinikum rechts der Isar, Germany*

**Editors:** Accepted in MIDL 2020

## Abstract

Robust localization of organs in computed tomography scans is a constant pre-processing requirement for organ-specific image retrieval, radiotherapy planning, and interventional image analysis. In contrast to current solutions based on exhaustive search or region proposals, which require large amounts of annotated data, we propose a deep reinforcement learning approach for organ localization in CT. In this work, an artificial agent is actively self-taught to localize organs in CT by learning from its asserts and mistakes. Within the context of reinforcement learning, we propose a novel set of actions tailored for organ localization in CT. Our method can use as a plug-and-play module for localizing any organ of interest. We evaluate the proposed solution on the public VISCERAL dataset containing CT scans with varying fields of view and multiple organs. We achieved an overall intersection over union of 0.63, an absolute median wall distance of 2.25 mm and a median distance between centroids of 3.65 mm.

**Keywords:** Organ localization, deep reinforcement learning, computed tomography.

## 1. Introduction

Within medical image analysis, the task of detecting and localizing organs or anatomical structures is a crucial pre-processing step towards robust and accurate quantitative analysis for diagnosis, treatment, and follow-up. For instance, image segmentation algorithms based on deep learning can be substantially improved by providing only the organ-of-interest to the segmentation algorithm, instead of processing in the full 3D volume. This pre-processing step reduces false-positive segmentations and helps leverage the computational resources efficiently. Automatic organ localization could also improve lesion detection, reducing the search space. Similarly, localizing an organ of interest in pre- and post-operative images can increase the overall performance of image-to-image registration in radiotherapy. Furthermore, an automatic organ localization is desired because manual localization is time-consuming and is prone to errors. Therefore, this work aims to automatically localize organs in CT images. This is a challenging task for computer-aided systems, as they are susceptible to variation in organs' shapes, sizes, fields of view and pathology among subjects.

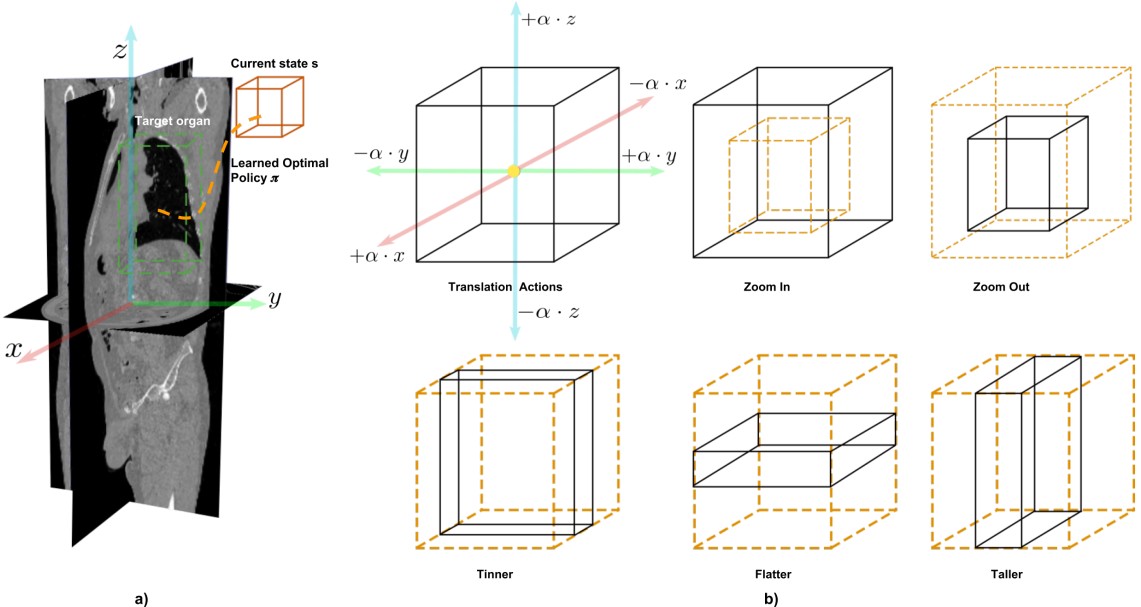

Figure 1: RL environment and action space for organ localization: a) The environment is the 3D CT scan. The agent is represented by the orange cube and the target organ in green. The optimal policy is learned during training. b) Schematic illustration of the action space, six actions for translation, two for scaling the whole box, and three actions to scale the box in each of the three directions.

Previous attempts for organ localization include multi-atlas registration (Jimenez-del Toro et al., 2016), and machine learning approaches with hand-crafted features (Criminisi et al., 2013; Pauly et al., 2011; Zhou et al., 2012). Although multi-atlas approaches perform well, they are computationally expensive. Similarly, machine learning methods are considered highly dependent on parameter tuning and feature selection. Current state-of-the-art organ localization approaches are based on deep-learning. In such approaches, either 2D CNNs are leveraged to solve a multivariate regression problem (Hussain et al., 2017; de Vos et al., 2017) or 3D object localization with object proposals (Xu et al., 2019) is studied. Nevertheless, 2D-3D CNNs require a large corpus of annotated training data, making them prohibitive for medical applications. Given that reinforcement learning (RL) has proven to be a successful approach for object detection in natural images (Caicedo and Lazebnik, 2015), as well as for landmark and lesion detection (Ghesu et al., 2017; Alansary et al., 2019; Maicas et al., 2017). In this work, we explore the capabilities of RL for organ localization. Notice the difference between previous works using RL for landmark detection and our approach for organ localization. In landmark detection the aim is to find a coordinate in the 3D space which describes the landmark.A fixed size box and translation actions are sufficient to find landmarks in the 3D space as described in (Ghesu et al., 2017; Alansary et al., 2019). In contrast, in organ localization we aim to predict a 3D bounding box that encompasses the organ of interest. Therefore, the deformation of the 3D box naturally

calls for other actions that can change the aspect ratio of the box to fit the organ of interest.

The proposed approach, based on deep Q-learning algorithm (Mnih et al., 2015) introduces an artificial agent that is taught to learn from its own experiences to find an optimal policy to localize the organ of interest. To this end, the artificial agent's task consists of sequentially deforming a bounding box to fit an organ, given a set of allowed actions, Fig 1a.

Specifically, our contributions in this work are the following:

- To the best of our knowledge, this is the first work for organ localization leveraging deep reinforcement learning.

- A crucial contribution to the success of our approach is the introduction of a new set of 11 actions, which are tailored for organ localization in RL to account for the variability of organs' sizes and shapes.

- We show that for the task of organ localization, RL can learn under a limited data regimen compared to CNNs.

## 2. Background

Reinforcement learning is a learning method inspired by cognitive science and animal behavior. It can be described as an artificial agent interacting with an environment $\mathcal{E}$ through a sequence of observations, actions, and rewards. At every time-step, the agent observes the current state $s$, and selects an action $a \in A$, where $A$ is a finite set of legal actions that the agent can take. The environment then emits a signal $R$, which is received by the agent and interpreted as the obtained reward for taking the action $a$. The final goal of the agent is to find an optimal behavior function that maximizes the future reward. This problem is formulated as a *Markov Decision Process* (MDP). Since the MDP is usually not fully observable, RL can be used to approximate the optimal function by iteratively sampling a set of states, actions, and rewards. Given the success of Q-learning as a surrogate to the optimal function in object detection (Caicedo and Lazebnik, 2015), as well as for landmark and lesion detection (Ghesu et al., 2017; Alansary et al., 2019; Maicas et al., 2017), we adapt the Q-learning strategy for the task of organ localization.

**Deep Q-Learning:** The optimal policy $\pi^*$ dictating the behavior of the agent, can be defined as an action-value function $Q^*(s, a) = \max_{\pi} \mathbb{E}[R_t \mid s_t = s, a_t = a, \pi]$ mapping states to actions by maximizing the expected reward $R_t$ following a policy $\pi$. Rewards are discounted by a factor $\gamma \in [0, 1]$ to weight immediate and future rewards. The expected future reward, is then defined as $\mathbb{E}[r_t + \gamma\, r_{t+1} + \gamma^2\, r_{t+2} + ... \mid s_t = s, a_t = a, \pi]$, where $t$ is the current time-step. The optimal value function obeys the Bellman equation, stating that if the optimal value $Q^*(s', a')$ of the next state $s'$ is known for all possible actions $a'$, then the optimal behavior is to select the action $a'$ that maximizes the expected value $r + \gamma Q^*(s', a')$. The action-value function can then be estimated by using the Bellman optimality equation in a recursive manner:

$$Q_{i+1}(s, a) = \mathbb{E}[r + \gamma\max_{a'} Q_i(s', a') \mid s, a] \tag{1}$$

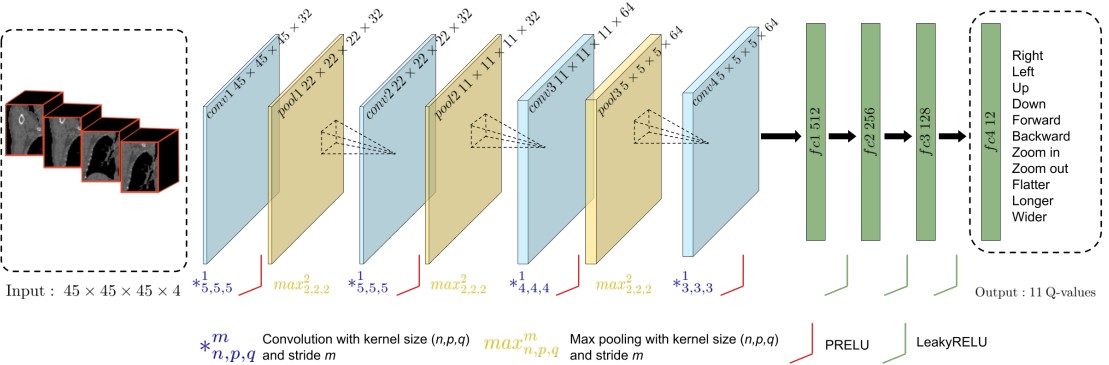

Figure 2: Network architecture for multi-organ localization. The input to the network is the current box voxel values plus the last 4 states. The output has dimension 11, which corresponds to the q-values for every action.

here $s'$ and $a'$ are the next state and action, respectively. For practical problems where the space set $S$ is too complex to explore, a non-linear function as deep neural network $Q^*(s, a, \theta)$ with parameters $\theta$ can be used as an approximation, resulting in the deep Q-learning algorithm (Mnih et al., 2015). During learning, at every time-step, an experience replay buffer $D$ stores the agent's experiences $e_t = (s, a, r, s', a')$, and mini-batches are uniformly sampled from it to optimize the network parameters by minimizing the loss function:

$$L_i(\theta_i) = \mathbb{E}_{(s,a,r,s',a') \sim U(D)}\Big[\big(r + \gamma \max_{a'} Q(s', a'; \theta_i^-) - Q(s, a; \theta_i)\big)^2\Big] \tag{2}$$

where $\theta_i$ are the network parameters at iteration $i$. $\theta_i^-$ are the parameters of the target network, which are updated only every $C$ iterations. Using the target network stabilizes the rapid policy changes. The optimization problem is then to minimize the squared difference between the target values and the q-values.

## 3. Method

To tackle the organ localization task, we define the problem as an MDP as described before. Formally, the MDP has a set of actions $A$, a set of states $S$, and a reward function $R$. This section defines in detail every component of the MDP.

**State**: Let the environment $\mathcal{E}$ be the 3D CT scan, then, every possible state $s$ in $\mathcal{E}$ is represented with the voxel values contained inside the current bounding box $b = [b_x, b_y, b_z, b_w, b_h, b_d] \in \mathbb{R}^6$, where $b_x, b_y, b_z$ denotes the top-left-front corner and $b_w, b_h, b_d$ the lower-right-back corner coordinates of the box. To stabilize the agent search and inform the agent about actions chosen in the past, the last 4 states are given to the network. Notice that, although our action space is discrete, its effect on the state space is continuous due to the $\alpha$ factor

described in the next lines. The position of the box is continuous in $\mathbb{R}^6$. It is only when extracting the intensity values in the CT scan that we discretize the box to work in the voxel space.

**Localization actions:** We define a new discrete set of legal actions $\mathcal{A} = \{t_x^+, t_x^-, t_y^+, t_y^-, t_z^+, t_z^-, s^+, s^-, d_x, d_y, d_z\} \in \mathbb{R}^{11}$ for organ localization. These actions enable the agent to reach every location in the environment and to consider all kinds of organ's shapes and sizes. The first six actions correspond to positive and negative translation in the $x, y, z$ directions. Notice that these six translation actions do not change the neither the size nor the aspect ratio of the box, rather only the position of the box in $\mathbb{R}^6$. The scaling actions zoom in $s^+$ and zoom out $s^-$ can be seen as a global scaling of the box. These actions will change the size of the box but preserve the aspect ratio. The actions thinner $d_x$, flatter $d_y$, and taller $d_z$ represent a deformation on one of the faces of the bounding box. These actions are responsible for changes in the aspect ratio of the box. Without these actions, the aspect ratio of the box will stay fixed from the initialization until convergence. The action set $\mathcal{A}$ is illustrated in Figure 1b. Any of the transformation actions make a discrete change of the box by a factor $\alpha \in [0,1]$ relative to its current size. The transformed box is then obtained by adding or subtracting $\alpha_h, \alpha_w, \alpha_z$ to the box coordinates. For instance, a positive translation of the box in the $x$ direction, can be performed by changing the box coordinates $b_x = b_x + \alpha_w$ and $b_w = b_w + \alpha_w$, with the relative factor $\alpha_w = \alpha * (b_w - b_x)$.

**Reward function:** The intersection over union (IoU) is the signal driving the agent. It measures the quality of taking an action $a$ at every time-step. Let $IoU(g, b)$ be the intersection over union between the target box $g = [b_x, b_y, b_z, b_w, b_h, b_d] \in \mathbb{R}^6$ and the predicted box. When the agent moves from state $s$ having box $b$ ends up in the next state $s'$ with box $b'$, the reward function is computed with:

$$R_a(s, s') = sign(IoU(b', g) - IoU(b, g)) \tag{3}$$

The reward implicitly tells the agent if the taken action $a$ improves the IoU. The reward is binary $r \in \{-1, +1\}$ and this quantification helps the agent to differentiate between good an bad actions at every step. Directly using the intersection-over-union (IoU) can lead to very small rewards towards convergence as the change in IoU will be smaller, almost insignificant as shown in (Caicedo and Lazebnik, 2015).

**End of Sequence:** The termination of the localization process is different for training and testing. During training, the agent stops the search when the IoU between the current state and the target coordinates is greater or equal to a predefined threshold $\tau$. During testing, we terminate the localization process when oscillation occurs as proposed in (Alansary et al., 2019).

**Network Architecture:** We adopted the network architecture from (Alansary et al., 2019) and tailored it for the organ localization task. We chose this architecture for its success in landmark detection. The input to the network consist of the voxel values of the last 4 steps that the agent has taken. The output of the network is a distribution over actions, shown in Figure 2.

|              | Avg IoU | Wall dist [mm]    | Centroid dist [mm] |
|--------------|---------|-------------------|--------------------|
| **Right Lung**   | 0.77    | $3.46 \pm 5.28$   | $6.06 \pm 10.25$   |
| **Left Lung**    | 0.73    | $4.91 \pm 7.38$   | $10.32 \pm 17.09$  |
| **Right Kidney** | 0.60    | $2.96 \pm 2.91$   | $5.69 \pm 5.67$    |
| **Left Kidney**  | 0.57    | $4.06 \pm 4.98$   | $7.52 \pm 9.02$    |
| **Liver**        | 0.80    | $2.41 \pm 0.70$   | $3.36 \pm 1.34$    |
| **Spleen**       | 0.60    | $5.25 \pm 7.23$   | $9.20 \pm 12.03$   |
| **Pancreas**     | 0.32    | $12.26 \pm 13.60$ | $20.79 \pm 20.38$  |
| **Global**       | 0.63    | $5.04 \pm 6.01$   | $8.99 \pm 10.82$   |
| **Median**       | 0.60    | 2.25              | 3.65               |

Table 1: **Quantitative results**: for every organ, the average IoU, the distance between the box walls and the distance between centroids is described. Global localization performance is also reported.

## 4. Experiments and Results

To demonstrate the performance of the proposed approach for organ localization, we test the localization algorithm in 7 different organs: right lung, left lung, right kidney, left kidney, liver, spleen, and pancreas.

**Dataset:** The dataset used in the experiments consists of CT scans from the VISCERAL dataset (Jimenez-del Toro et al., 2016) for multi-organ segmentation. The scans include CTs with and without contrast enhancement (ceCT). Two different fields of view; whole-body and thorax are present in the dataset. In order to show that the agent can learn from limited annotated data, we use 70 CT scans for training and 20 for testing. Each volume is isotropically resampled to $3mm^3$ resolution and normalized to zero mean and unit variance.

**Evaluation Metrics:** To evaluate the accuracy of the proposed RL-localization approach, we report the IoU between the ground truth location $g$ and the predicted box $b$. The widely used absolute wall distance, and the centroid distance between the predicted box and the ground truth box are also included. All the wall and centroid distances are reported as mean $\pm$ standard deviation. Median values are also reported to account for outliers.

**Training localization agent:** During training, the agent follows an $\epsilon$-greedy policy, gradually shifting from exploration to exploitation as training takes place. To encourage the agent to visit only states that maximize the future reward and therefore, reduce the search space, we use guided exploration proposed in (Caicedo and Lazebnik, 2015). In guided exploration, at every step, only actions returning positive reward can be randomly chosen.

We train one artificial agent per organ for 30 epochs. The first 20 epochs are annealed from 1 to 0.1 following the $\epsilon$-greedy strategy. This allows the agent to learn gradually from its own model. During training, the threshold to stop the localization process $\tau$ is set to 0.85. When the IoU is greater or equal to 0.85, the organ of interest is considered as localized and the agent restarts in a new CT scan. For testing, we stop the localization process when oscillation occurs. Finally, $\alpha$ dictating the transformation factor for every action is set to 0.1. This value is found empirically, smaller values of $\alpha$ result in slower convergence while higher values for $\alpha$ reduce the localization performance. Since $\alpha$ plays an important

| Method | | Organs | | | | | | | Time (s) |
|---|---|---|---|---|---|---|---|---|---|
| | # Scans | L Lung | R Lung | L Kidney | R Kidney | Liver | Spleen | Pancreas | |
| RF (Criminisi et al., 2013) | 400 | 12.90 | 10.10 | 13.60 | 16.10 | 15.70 | 15.50 | - | 4 |
| RF (Gauriau et al., 2015) | 130 | - | - | 5.50 | 5.60 | 10.70 | 7.90 | - | 3.2 |
| RF (Samarakoon et al., 2017) | 100 | - | - | 11.52 | 10.98 | 15.82 | 14.84 | - | 2.2 |
| CNNs (Mamani et al., 2017) | 553 | 2.87 | 2.60 | 5.68 | 5.82 | 8.19 | 7.17 | - | - |
| CNNs (Humpire-Mamani et al., 2018) | 1884 | **2.31** | **1.99** | **2.67** | 3.03 | 5.84 | **3.37** | - | 4.0 |
| 3D RCNN (Xu et al., 2019) | 118 | 5.1 | 4.9 | 4.3 | 3.9 | 8.5 | 6.3 | **9.2** | **0.3** |
| **Ours (100% data) RL** | 70 | 4.91 | 3.46 | 4.06 | **2.96** | **2.41** | 5.25 | 12.26 | 3.1 |
| **Ours (10% data) RL** | **7** | 8.28 | 7.90 | 9.25 | 6.60 | 6.16 | 7.91 | 17.83 | 3.1 |

Table 2: **Comparison with other methods**: the table describes the mean wall distance in mm for each algorithm. Note that the results of other algorithms are obtained on different data-sets from the original publications. The first three methods correspond to random forest (RF) approaches. The subsequent three methods leverage CNNs and a large number of annotated scans. Finally, our method based on reinforcement learning (RL) is described in the last two rows of the table.

role in the proposed approach, we believe that exploring this hyper-parameter could lead interesting directions. For instance, annealing the $\alpha$ value can help for faster convergence. Using a multi-scale extension where the value of $\alpha$ starts with a big value (scale) and after convergence, reduce the scale could potentially lead to improvements in localization performance. Notice the training strategy and the final performance is independent of the organ of interest. It is the hyper-parameter $\alpha$ that can be customized depending on the size of the organ to localize. In this work, this hyper-parameter is fixed. The study of this parameter is considered for future work.

Training an agent for organ localization takes 13-20 hours depending on the number of steps required by each organ's agent to reach its target. The networks are trained in a NVIDIA Titan Xp GPU, with a batch size of 48 and experience replay of $14 \times 10^3$.

**Evaluating the localization performance:** Table 1 describes the quantitative evaluation for every organ, as well as the overall performance. In our experiments, we observed that bigger organs such as lungs and liver yield better localization, while the performance of smaller organs (kidneys, spleen) is compromised. We attribute this issue to the fixed step size $\alpha = 0.1$ that the agent can take in every time-step. Setting organ-specific $\alpha$ values could improve the localization performance. In our case, smaller values of $\alpha$ make the agent converge slower, and bigger values decrease the localization performance. We obtained a median absolute distance of 2.35 mm for the walls and 3.36 mm for the centroids.

Moreover, to show the effectiveness of our RL approach to learn under low data regiments, we performed an ablation study. We trained the proposed approach using only 10 % of the total training data (7 scans) as shown in the last row of Table 2. We observed that the wall distance decreased by half compared to the system training with 100 % of the data. Although the performance of our localization approach declines when reducing the training data, our system is able to match or outperform those systems trained with

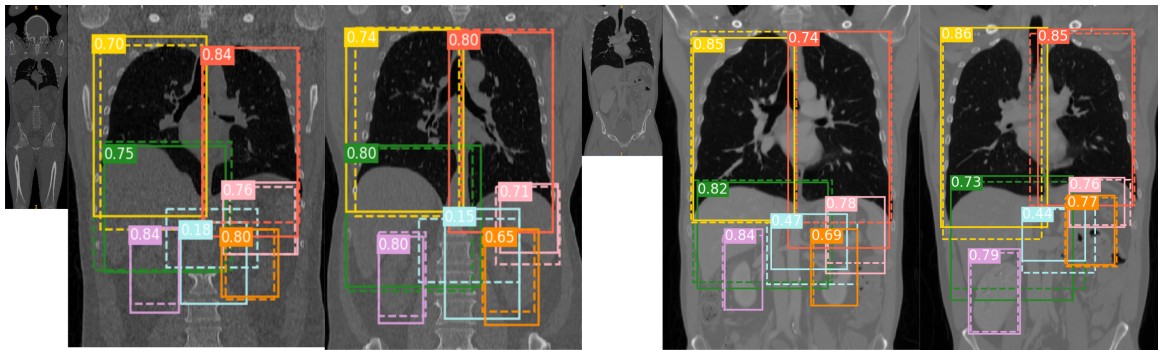

Figure 3: **Qualitative results**: Each image shows the mid-slice in the coronal view. For each organ, the dotted lines correspond to the ground truth location, while solid lines denote the agents' prediction. The small image represents the field of view of the adjacent sample images. The first set of images is whole-body CT without contrast, and the last set of images is thorax ceCT. Each organ' truth location and network prediction is illustrated with a different color. Liver ●, right lung ●, left lung ●, right kidney ●, left kidney ●, spleen ●, and pancreas ●. IoU is displayed for every organ. (Visualized better in color)

hundreds of annotated scans. We attribute this to the fact that in the proposed work, RL learns a mapping from states to actions similar to the way a human would localize organs with a visualization tool. Rather than regressing the coordinates of the box comprehending the organ of interest, the human strategy would be to find the slices that contain the organ in every orthogonal view, mark down the bounding box and later refine the selected box. Furthermore, in our environment, there are in theory infinite number of paths to the target organ. This is translated in an infinite number of training samples to learn from, resulting on RL being able to learn under low data constrains.

**Comparison with other methods:** To evaluate the performance of our method in comparison with previous approaches, we included a comparison to machine learning-based methods (Criminisi et al., 2013; Gauriau et al., 2015; Samarakoon et al., 2017), deep learning-based methods working in 2D (Mamani et al., 2017; Humpire-Mamani et al., 2018) and deep-learning methods leveraging 3D information (Xu et al., 2019) illustrated in Table 2. Given that there is no benchmark for organ localization in CT, the numbers reported in Table 2 are obtained from the authors' reports, similar to previously reported results (Humpire-Mamani et al., 2018; Xu et al., 2019). The mean wall distance per organ is reported for comparison purposes. We can observe that (Humpire-Mamani et al., 2018) have the overall best performance. It achieves the lowest wall distance in the majority of evaluated organs. Nevertheless, it uses a large number of annotated scans ($\sim$ 2000) for training. This makes the organ localization scale-prohibitive. To be able to achieve the described performance in other organs, a large number of annotated CT scans are needed. Concerning the inference time, the proposed method in (Xu et al., 2019) yields the lowest inference time.

**Visualizing the localization:** To qualitatively evaluate the organ localization, Figure 3 shows sample volumes in the testing dataset with its corresponding organ localization. Dotted lines describe the ground truth locations and solid lines the predicted box. Each organ is represented with different color as described in Figure 3. We can observe that the artificial agent is able to localize most of the organs with IoU values between 0.65 to 0.9. In most of the cases the agent either includes all the organ of interest in the prediction, or the predicted box is close to the ground truth. This is a desired property for algorithms that use localization as as a pre-processing step.

Finally, we state that the value of the proposed organ localization approach with RL compared to supervised learning is that RL does not rely on a large corpus of annotated data. With the proposed approach, we are able to learn the given task by training on a relatively small dataset (70 scans) compared to deep supervised learning approaches, which would have needed hundreds of training example to successfully localize organs. More specifically, notice that (Humpire-Mamani et al., 2018) uses 1884 to achieve the highest performance as seen in Table 2 . This is attributed to the fact that RL does not depend on image-annotation pairs. Instead, the agent is taught to solve a task in sequential steps in an optimal way. Every step towards the target represents a training example. Therefore, from one single CT scan, the agent can take hundreds of training samples. Since there are in theory infinite paths towards a solution, there are also vast learning samples.

## 5. Conclusion

In this work, we explore, for the first time, the organ localization task with deep reinforcement learning. We introduced a new set of actions, tailored for organ localization. We demonstrate that RL can be effective for training in limited data regimes as opposed to supervised learning approaches, which need $\sim 20$ more times the data. Finally, we evaluated our approach for the localization of seven organs with diverse shapes, sizes, appearances as well as different fields of view, showing the effectiveness of RL for organ localization.

In future work, we plan to combine multi-agent deep reinforcement learning as well as multi-scale organ localization. The latter is of especial interest for small organ localization.

## Acknowledgments

The authors gratefully acknowledge the Deutsche Forschungsgemeinschaft (DFG, German Research Foundation) - GRK 2274 for the financial support. The authors thank the support of NVIDIA Corporation with the donation of the Titan Xp GPU used for this research.

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
