# OpenReview forum: "Deep Reinforcement Learning for Organ Localization in CT"
_MIDL.io/2020/Conference — MIDL 2020_

### Official Review · AnonReviewer1 · 2020-02-20
**well written paper with little proven methodological novelty**

**Rating:** 3
**Confidence:** 5
**Recommendation:** Oral

**Summary:**

This paper proposed to use RL to localise organs in CT scans. The authors propose to introduce five new scaling actions for the agent's view.

Alansary et al. observed the the performance of different training strategies highly depends on the target. This has not been discussed in the presented work.
Also, the proposed method would end up with target-specific agents. Would it be possible to link the training and exploration of the task-specific agents to a multi-agent system through sharing their CNN weights? This has been proposed in previous work to make training and inference more robust and accurate and it should be included in this work (not only as future work in the very last sentence).

Overall, the paper is well written and presents good results. However, methodologically it is almost indistinguishable from previous work in this domain. The contribution of the proposed scaling actions has not been thoroughly studied. An ablation study has been done with respect to the number of training samples but equally important would be to study the agents' performance when using only six translation actions instead of the proposed 11. Hence, the only novelty can be found in the application of RL to CT organ localisation.

**Strengths:**

- application of RL is relevant and can lead to alternative research directions
- the paper is well written
- the comparison to other CT organ localisation work has been done well
- it has been shown that RL agents can learn from little data

**Weaknesses:**

- the ablation study is incomplete and would also need to investigate the contribution of the scaling actions
- multi-agent systems would need to be compared to rpoperly base on state-of-the-art
- It's only the 'first-time' that RL has been evaluated for this very particular niche task

**Detailed Comments:**

see above

**Justification Of Rating:**

The area and application is interesting and if the authors manage to address my concerns it should be an interesting oral presentation at MIDL.



Please remove this minimum character requirement. What else shall I say? I have been personally invited to review for MIDL so I should have competence to do so. A paper-matching system is in place to make sure the reviewers match the paper topic, so what else shall I justify here?

**Paper Type:**

validation/application paper

**Questions To Address In The Rebuttal:**

- how do the agents perform with only six actions
- how would a multi-agent system with shared weights perform
- how does the training strategy impact performance, especially per organ?

**Special Issue:**

yes

---

> ### Author Response · Authors · 2020-03-27
> **Response to AnonReviewer1**
>
> Thanks for your valuable feedback and your recommendation for special issues. We will try to address your comments as detailed as possible
>
> C1: The ablation study is incomplete and would also need to investigate the contribution of the scaling actions. How do the agents perform with only six actions?
>
> It is unclear what the reviewer means by “scaling actions”: 1) if you refer to all actions expect translation, i.e: zoom in, zoom out, thinner, flatter and taller, then the answer to your question is that this actions are crucial for organ localization and therefore part of the contribution. Without these actions, the agent needs a heuristically fixed box size per organ. This would decrease the localization performance. 2) if your question is regarding zoom in and zoom out. These actions can be seen as a global scaling of the box. These actions conserve the aspect ratio of the box. 3) if your question is regarding the thinner, flatter and taller. These actions are responsible for changes in the aspect ratio of the box. Without these actions, the aspect ratio of the box will stay fixed from the initialization until convergence.
>
> We have edited the localization actions section on Page 4 to better illustrate the contribution of each action in the action set. Especially the scaling actions.
>
> C2:  Multi-agent systems would need to be compared to properly base on state-of-the-art. How would a multi-agent system with shared weights perform
>
> To the best of our knowledge, there is no prior work using multi-agent for organ localization. Our approach is the first using deep reinforcement learning for this task. While multi-agent systems have been proposed for landmark detection (Vlontzos et. Al 2019), there is no literature on this regard for organ localization. Thus, it cannot be considered for comparison. In case we missed a relevant prior work, we request the reviewer to point us to it.
>
> Vlontzos, A., Alansary, A., Kamnitsas, K., Rueckert, D. and Kainz, B., 2019, October. Multiple Landmark Detection using Multi-Agent Reinforcement Learning. In International Conference on Medical Image Computing and Computer-Assisted Intervention (pp. 262-270). Springer, Cham.
>
>
> C3: How does the training strategy impact performance, especially per organ
>
> In our approach, the training procedure is agnostic to organ type. Therefore, it is extendable to any anatomical structure. We also added additional training details in the “Training localization organ” section to provide better intuition on the chosen scheme.

---

> > ### Comment · AnonReviewer1 · 2020-03-29
> > **borderline reply. I recommend Weak accept.**
> >
> > I mean 'zoom', which is equal to scale. Removing this in the ablation study would show how much it actually adds to the overall approach.
> > you can compare localisation accuracy in mm to multi-agent landmark detection approaches. These tasks are not so different from each other.

---

> > > ### Author Response · Authors · 2020-04-01
> > > **Response to AnonReviewer1 Action space**
> > >
> > > Indeed to be able to asses the contribution of each action, one action at a time should be excluded.
> > > We found the actions space empirically but due to space restrictions, we did not contributions from each action. We observe that the effect "zoom" is a faster convergence. The agent may start the search from a relatively big box with respect to the organ of interest. Thus, the "zoom" actions make the agent converge faster.
> > >
> > > To compare to landmarks literature, different issues can appear that may introduce unwanted biases in one part or the other. For example, which point of the organ box do we consider as a landmark. The landmark can be different for every organ box across patients. Therefore not a direct comparison is possible. We would need to have manual annotations for landmarks on the dataset used for organ localization for a fair comparison.

---

### Official Review · AnonReviewer3 · 2020-03-13
**an extension of the Alansary et al. (2019) RL landmark localization paper on the organ localization**

**Rating:** 3
**Confidence:** 4
**Recommendation:** Poster

**Summary:**

Although the authors presented the first RL approach for organ localization, their work has limited technical contribution as being an extension of the RL based approach for landmark localization proposed by Alansary et al., 2019. The modification made by the authors is in the output of the Q network where, in addition to the six actions (left, right, up, down, forward) used in the landmark localization, there are now five new ones (zoom in, zoom out, flatter, longer, wider). The method is evaluated on VISCERAL data set and the results are in-line with SOTA.

**Strengths:**

(1) The authors have shown that reinforcement learning can also be used on the task of organ localization.
(2) The method is evaluated on a publicly available dataset.
(3) The method is clearly presented.


**Weaknesses:**

In addition to the lack of technical contribution, my main concern is the evaluation of the method. Although the authors used a publicly available dataset (VISCERAL), none of the methods they compare to has been evaluated on this dataset. On the other hand, Xu et al. (2019) made the annotation of the LiTS dataset as well as their code publicly available for comparison. Thus, authors were able to directly compare with Xu et al. using the same dataset or running the available code on the VISCERAL dataset. Moreover, the authors used only one split for their method evaluation (70 images for training and 20 CT images for testing) which is not the best strategy for evaluating the model performance on a limited dataset. Authors should use k-folds cross-validation.

Due to the above mentioned, a direct comparison of the methods is not possible. However, if we just compare the numbers, the proposed RL method only clearly outperforms RF-based methods, whereas, in comparison to the CNN methods, is at best in-line. This is probably why the authors decided to go for the experiment with a limited number of training images. However, the experiment with only seven training images is not clearly explained. How were the seven images selected, did authors performed cross-validation? Moreover, other methods have not been evaluated on a reduced number of training images, so no comparison can be made.

Finally, the authors should not claim that CNN methods “would have needed hundreds of training examples to successfully localize organs”, since e.g. Xu et al. method used 118 images (compare to 70 used in this work) and achieved the same localization results.

**Justification Of Rating:**

The RL is novel direction in MIA community that has not been evaluated on the task of organ localization. The presented results are in-line with SOTA, although they were not evaluated on the same dataset.

**Paper Type:**

methodological development

**Special Issue:**

no

---

> ### Author Response · Authors · 2020-03-27
> **Response to AnonReviewer3**
>
> We thank reviewer 3 for their constructive comments and valuable feedback. We will try to address your questions and suggestions below:
>
> C1: The major concern is regarding the validation strategy.
>
> Benchmarking organ localization has always suffered from a lack of standardization. This is due to usage of private dataset and not releasing code (Criminisi et al., 2013; Pauly et al., 2011; Zhou et al., 2012, Hussain et al., 2017; de Vos et al., 2017, Humpire-Mamani et al., 2018). We report our performance along similar lines of previous work. Furthermore, other works such as Xu et.al (2019) released their code in a hermetic configuration, requiring specific computer system and packages which are not open source. See all the open issues in [1] for reference. Moreover, implementing previous work code can introduce unwanted bias in one part or another.
> For the follow-up work, we are in conversation with the authors of prior works to standardize performance.
>
> [1] https://github.com/superxuang/caffe_3d_faster_rcnn/issues
>
> C2: Authors should use k-folds cross-validation.
>
> We agree that cross-validation should be done. We could not do it because of time constraints. However, notice that we do not draw major conclusions on performance superiority. We rather focus on showing an alternative solution for organ localization and showing how a deep-reinforcement learning approach can overcome current challenges in this task: agnostic to the field of view, to image modality (CT and contrast CT) and to limited data constrains. We propose an alternative approach towards obtaining this robustness as the artificial agent learns the human anatomy rather than low-level image features.
>
> C3: the authors should not claim that CNN methods “would have needed hundreds of training examples to successfully localize organs”
>
> We apologize for the generic claim. Our intention was in reference to (Humpire-Mamani et al., 2018) which uses 1884 scans as shown in Table 2. We have rewritten this part.

---

> > ### Comment · AnonReviewer3 · 2020-04-03
> > **not convincing response, but I'm staying with borderline accept.**
> >
> > I haven't compiled the Xu et.al (2019) method, but after checking the requirements and the issues others had, I would not say the code is in "a hermetic configuration". It might take some time to get it running but it is definitely doable.
> >
> > If the authors did not have time to perform the appropriate experimental validation, they should have waited until the next MIA conference to submit the manuscript. MICCAI was only a month later.

---

### Official Review · AnonReviewer4 · 2020-03-19
**Reinforcement Learning for Localization**

**Rating:** 3
**Confidence:** 5
**Recommendation:** Poster

**Summary:**

The paper proposes to use reinforcement learning (Deep Q-Learning) to find the correct slice to localize a given organ in CT. It claims to be the first work for organ localization with RL and the results are promising in comparison with other non-RL methods, especially in scarse data scenarios, tested on multiple organs.

**Strengths:**

- The work is carried out on a public dataset. If the authors release the code, reproducability of the results would add value.
- Comparison with other methods is presented
- The method is sound and seem to work fine
- Evaluation is thorough and draws a clear picture.

**Weaknesses:**

- Using a discrete deep RL method to solve a problem which naturally calls for a continuous action space.
- Given the discrete action formulation, the authors seem to have missed to disclose the size of the translation steps (t) taken at each step and how annealing this value might have improved the results
- The paper describes the Deep Q-Learning (DQL) algorithm in much detail; however, given the history of the method, the details could have been minimized.
- This is not the first plane localization using RL in the medical imaging domain. There are similar works such as https://arxiv.org/pdf/1806.03228.pdf who also use DQL for localization.

**Justification Of Rating:**

The authors seem to replicate the work of [Alansary, 2019] on landmark detection by changing the problem to organ localization. However, there have been other attempts to use RL for plane localization (and with the exact same RL method) which robs the paper of the novelty in application.

**Paper Type:**

validation/application paper

**Questions To Address In The Rebuttal:**

- Why not use a Deep RL method applicable to the continuous domain?
- It would be interesting to show the cases where the agent failed to find the correct plane. If so, why do you think the agent was failing in those cases?

**Special Issue:**

no

---

> ### Author Response · Authors · 2020-03-27
> **Response to AnonReviewer4**
>
> We thank the reviewer for the extensive comments as well as constructive suggestions, we indeed plan to release the code for reproducibility. We will try to address the raised concerns:
>
> C1: Using a discrete deep RL method to solve a problem that naturally calls for a continuous action space.
>
> Our action space is discrete. Nevertheless, its effect on the state space is continuous due to the ‘alpha’ component. The position of the box is continuous in Rˆ6. For example, consider the case of translation.  If the box size is 121ˆ3 and alpha=0.1. The agent translates the box 12.1 mm, the new position of the box will be in the continuous space. It is only when extracting the intensity values in the CT scan that we discretize the box to work in the voxel space. We have made this more explicit in section “State” on page 4.
>
> We thank the reviewer for the idea of continuous action spaces along the lines of an actor-critic extension. We will explore this idea further.
>
> C2: Given the discrete action formulation, the authors seem to have missed disclosing the size of the translation steps (t) taken at each step and how annealing this value might have improved the results
>
> On page 6, section “Training localization agent”, we describe the step size alpha. Annealing the alpha value will impose another hyperparameter to tune and we do not study this effect in this work. We believe that annealing the alpha value can help for faster convergence but would have a marginal effect on the localization performance.
> The observation of the reviewer is the lines of a multi-scale extension of our approach and it is considered in future work.
>
> We have extended the explanation of the effect of alpha on page 6 section “Training localization agent” for the revised version.
>
> C3: The paper describes the Deep Q-Learning (DQL) algorithm in much detail; however, given the history of the method, the details could have been minimized.
>
> The intention of an extensive description of Deep Q-Learning was to create a self-contained paper. We wanted that readers interested in our approach for organ localization but not familiar with DQL paradigm have a clear understanding of our method.
>
> We thank the reviewer for calling attention to this detail. Detail description will be omitted in future work.
>
> C4: This is not the first plane localization using RL in the medical imaging domain. There are similar works such as https://arxiv.org/pdf/1806.03228.pdf who also use DQL for localization.
>
> Looking at organ localization from a view-planning perspective is an interesting idea. However, note that identifying the “best plane” is not sufficient for drawing a box around an organ.  For example, finding a plane that passes through the kidney does not warrant that this plane doesn’t contain other organs. Modifications to the action space in the mentioned work would be needed to enable the agent to perform organ localization. We thank the reviewer for offering us this perspective.

---

### Official Review · AnonReviewer2 · 2020-03-19
**An Interesting Idea on Classical Organ Detection Task in 3D CT.**

**Rating:** 2
**Confidence:** 5
**Recommendation:** Poster

**Summary:**

It is a new method to detect and localize organ bounding boxes within 3D CT using reinforcement learning (deep Q learning).
The reasonable action space and reward function are carefully designed for the application.
The experimental results are okay with necessary comparison with other baseline methods.

**Strengths:**

The paper is well-written and well organized.
Experimental results support the claim made in the paper.
The idea using reinforcement learning methods for bounding box detection in CT organ localization is relatively new.
The action space and reward function are carefully designed.
The paper presented a new application of reinforcement learning in medical image analysis.

**Weaknesses:**

The task of organ detection/localization has been studied for decades. The detection task itself is somehow simplified since each subject contains one major organ only. It would be interesting to see how the proposed method is applied on some challenging task, e.g. vertebra localization.
The performance of the proposed approach does not show clear advantage over previous state-of-the-art methods.

**Detailed Comments:**

The performance of (Ghesu et al., 2017) should be compared here since it shares the similar idea with the similar application.


**Justification Of Rating:**

Overall: the paper’s idea is interesting, but the performance of the proposed method has not shown clear advantage (both in accuracy and efficiency). The similarity of the proposed method is similar with the literature both in medical image analysis and computer vision.

**Paper Type:**

methodological development

**Questions To Address In The Rebuttal:**

What is the performance difference when the reward function returns a floating value?

**Special Issue:**

no

---

> ### Author Response · Authors · 2020-03-27
> **Response to AnonReviewer2**
>
> We appreciate your valuable review and helpful feedback. We will try to address your questions and suggestions below:
>
> C1: The performance of the proposed approach does not show a clear advantage over previous state-of-the-art methods. The performance of the proposed method has not shown clear advantage (both in accuracy and efficiency)
>
> We present our approach as a novel proof-of-concept introducing RL into an organ localization framework. We argue that RL makes more sense in the anatomical context. The artificial agent is enforced to learns the anatomy rather than other low-level features. Indeed, supervised models cannot guarantee this, for instance, it has demonstrated that CNNs are biased towards texture (Geirhos et. Al 2018). Our approach thus results in certain advantages over prior work based on supervised learning as described below:
>
> 1) Previous approaches train and test their localization algorithms in one field of view (FoV), while our approach is agnostic to the FoV. Our algorithm is trained in whole-body CT and thorax CT while state-of-the-art methods, for example, Xu et.al (2019) benchmark his algorithm using the LiTS dataset, which is highly standardized (one FoV).
> 2) Our approach uses different CT modalities during training, contrast CT and non-contrast CT, showing its generalizability and robustness.
> 3) We have also shown that our approach can learn under low data regimen constrains. The artificial agent can learn the localization task with only seven annotated scans without severe performance decrease. This is an interesting finding a clear difference to previous work.
>
> Increasing the accuracy and efficiency of the localization performance is considered in future work as described in the conclusion section. Multi-agent and multi-scale versions of the proposed approach are examples of them.
>
> Geirhos, R., Rubisch, P., Michaelis, C., Bethge, M., Wichmann, F.A. and Brendel, W., 2018. ImageNet-trained CNNs are biased towards texture; increasing shape bias improves accuracy and robustness. arXiv preprint arXiv:1811.12231.
>
> C2: The task of organ detection/localization has been studied for decades. The detection task itself is somehow simplified since each subject contains one major organ only. It would be interesting to see how the proposed method is applied to some challenging tasks, e.g. vertebra localization.
>
> The proposed approach is a single agent. We opine that extending it to a seemingly complicated task of vertebrae localization would not be successful. This is because of the similarity in appearance, shape, and texture within vertebrae. However, a hierarchical extension of our approach would work well.
>
> We believe that organ localization, although highly studied, suffers from robustness issues in terms of field of view, severe pathologies within the organ, and image modality (CT vs contrast CT). Our method is a step towards obtaining this robustness as the artificial agent learns the human anatomy rather than low-level image features.
>
> C3: The performance of (Ghesu et al., 2017) should be compared here since it shares a similar idea with a similar application.
>
> There exist fundamental differences between (Ghesu et al., 2017) and our approach, which does not warrant a direct comparison. Here we describe these differences.
>
> 1) Ghesu et al., 2017 perform landmark detection, they predict a coordinate in the 3D space (x,y,z), which does not have any anatomical significance. Our approach performs organ localization, predicts the 3D box that contains the organ of interest. This means predicting two coordinates (x,y,z) in the space.
> 2) We cannot compare points (x,y,z) vs boxes (x,y,x,w,h,d). For a direct comparison, (Ghesu et al., 2017) would have to define the landmarks as the centers of the bounding box. This is clearly not the case.
> 3) (Ghesu et al., 2017) uses the distance to the ground truth landmark as an evaluation metric. In our approach, we have intersection-over-union to the ground truth box and distance to the walls as evaluation metrics.
>
> We make these differences explicit in the introduction section.
>
> C4: What is the performance difference when the reward function returns a floating value
>
> With a floating point, we assume you are referring to the case of returning continuous rewards vs binary or discrete rewards. Directly using the intersection-over-union (IoU) can lead to very small rewards towards convergence as the change in IoU will be smaller. This is avoided by binarization as explained in (Caicedo and Lazebnik, 2015) in section 3.3. We use the same approach in our experiments.
>
> We now make this clear in the “reward function” setting.

---

### Meta-Review · Area_Chair1 · 2020-04-07
**MetaReview of Paper128 by AreaChair1**

**Rating:** 3
**Recommendation For Accepted Papers:** Poster

**Metareview:**

Overall, the reviewers find this work of enough interest to warrant acceptance. The method is an extension of prior work on reinforcement learning in medical imaging and any claims around this work being the "first time" should be rephrased to reflect this.

**Paper Type:**

validation/application paper

**Special Issue:**

no

---

> ### Author Response · Authors · 2020-04-09
> **Response to AreaChair1**
>
> We thank the Area Chair for summarizing the discussion around our work. We also thank the recommendation for acceptance.
>
> Here we address your concerns:
>
> “The method is an extension of prior work on reinforcement learning in medical imaging and any claims around this work being the "first time" should be rephrased to reflect this”.
>
> We want to emphasize that this is indeed the “first work for organ localization in CT using reinforcement learning”. We understand the confusion with previous approaches for landmark detection. The final version of the article will rephrase these claims to avoid confusion between organ localization and landmark detection.

---

### Decision · Program_Chairs · 2020-04-11

Accept